# Complex Chromosomal Rearrangement Involving Chromosomes 10 and 11, Accompanied by Two Adjacent 11p14.1p13 and 11p13p12 Deletions, Identified in a Patient with WAGR Syndrome

**DOI:** 10.3390/ijms242316923

**Published:** 2023-11-29

**Authors:** Andrey V. Marakhonov, Tatyana A. Vasilyeva, Marina E. Minzhenkova, Natella V. Sukhanova, Peter A. Sparber, Natalya A. Andreeva, Margarita V. Teleshova, Fatima K.-M. Baybagisova, Nadezhda V. Shilova, Sergey I. Kutsev, Rena A. Zinchenko

**Affiliations:** 1Research Centre for Medical Genetics, Moscow 115522, Russia; vasilyeva_debrie@mail.ru (T.A.V.); maramin@mail.ru (M.E.M.); natelasukhanova@gmail.com (N.V.S.); psparber93@gmail.com (P.A.S.); nvsh05@mail.ru (N.V.S.); kutsev@mail.ru (S.I.K.); renazinchenko@mail.ru (R.A.Z.); 2Dmitry Rogachev National Medical Research Center of Pediatric Hematology, Oncology and Immunology, Moscow 117997, Russia; andre180593@mail.ru (N.A.A.); teleshova_m@mail.ru (M.V.T.); 3Medical Center “Evromed”, Cherkessk 369000, Russia; baybagisova89@mail.ru

**Keywords:** WAGR syndrome, nephroblastoma, Wilms tumor, congenital aniridia, genetic testing, complex chromosomal rearrangement, *PAX6*, *WT1*, *LMO2*

## Abstract

Three years ago, our patient, at that time a 16-month-old boy, was discovered to have bilateral kidney lesions with a giant tumor in the right kidney. Chemotherapy and bilateral nephron-sparing surgery (NSS) for Wilms tumor with nephroblastomatosis was carried out. The patient also had eye affection, including glaucoma, eye enlargement, megalocornea, severe corneal swelling and opacity, complete aniridia, and nystagmus. The diagnosis of WAGR syndrome was suspected. De novo complex chromosomal rearrangement with balanced translocation t(10,11)(p15;p13) and a pericentric inversion inv(11)(p13q12), accompanied by two adjacent 11p14.1p13 and 11p13p12 deletions, were identified. Deletions are raised through the complex molecular mechanism of two subsequent rearrangements affecting chromosomes 11 and 10. WAGR syndrome diagnosis was clinically and molecularly confirmed, highlighting the necessity of comprehensive genetic testing in patients with congenital aniridia and/or WAGR syndrome.

## 1. Introduction

WAGR syndrome (OMIM #194070) is a rare chromosome syndrome that occurs in 1 child per 1,000,000 births [1]. WAGR syndrome patients present Wilms tumor (W), aniridia (A), genitourinary anomalies (G), and mental retardation (R). Out of the four classical WAGR syndrome features, aniridia is defined in all patients, Wilms tumor in 2/3, genitourinary anomalies in almost all male patients, and intellectual disability in more than 2/3 [2]. Additionally, other severe complications can be observed, such as glaucoma, cataracts, keratopathy, nystagmus, and other ocular manifestations; renal disease; cardiopulmonary, head, eyes, ears, nose, and throat anomalies; and behavioral and neurologic abnormalities [2]. The median age of renal tumor development is 19–23 months [2]. After the age of 5 years old, Wilms tumor development is much less probable. Usually, 11p13 interstitial deletions of contiguous genes region encompassing 18–62 genes, including at least the *PAX6* and *WT1* genes, serve as a cause of the syndrome [3]. Complex rearrangements involving more than 1 chromosome occur in 2 out of 34 patients with defined 11p13 chromosome deletions [4].

Here we report an extremely rare case of a patient with a giant early developed embryonic renal tumor, glaucoma, keratopathy, and complete bilateral aniridia associated with a complex chromosomal rearrangement, balanced translocation t(10,11)(p15;p13), pericentric inversion inv(11)(p13q12), and two adjacent deletions at the breakpoints of the complex chromosomal rearrangement.

## 2. Results

### 2.1. Case Presentation

A 3100 g, 48 cm Caucasian boy was born to a 29-year-old woman at 38 weeks of gestation. He was born with a loop of cord and fetal hypoxia in labor. Apgar scores were 8 and 9 at the 1 and 5 min. The patient was born in a second pregnancy in a non-consanguineous family. His parents are healthy. The infant was discharged with the diagnoses of congenital bilateral complete aniridia, hypospadias, cryptorchidism, and mild hypotonia.

At the age of 12 months, the patient referred to the ophthalmologist with complaints of corneal opacity, eye enlargement, and nystagmus. An ophthalmologic examination included corneal biomicroscopy, gonioscopy, intraocular pressure measurement, and retina examination. A fine horizontal nystagmus was noted. Corneal biomicroscopy revealed initial corneal pannus in the shape of a thin strip without vascularity, megalocornea, cornea swelling, complete aniridia, and glaucoma. The central corneal thickness was 720 µm and 664 µm in R/E and L/E, respectively (the normal range is 549 ± 46 µm). The retina examination revealed disc pallor, normal cup, and absence of the foveal reflex. Gonioscopy confirmed hypoplasia of the iris and mesodermal tissue rests in the anterior chamber angle. The intraocular pressure (IOP) measured with rebound tonometry (IC 200, iCare, Vantaa, Finland) was 27 mm Hg and 29 mm Hg in the R/E and L/E, respectively (the normal range is 16.08 ± 3.08 mm Hg). Since 12 months of age, the patient received initial treatment of topical anti-glaucoma medication with positive dynamics.

At the age of 13 months, a control ophthalmologist examination showed a positive effect of conservative therapy. The proband was visually rehabilitated with photochromatic glasses. Preservative-free artificial tears eye drops (carboxymethylcellulose) were prescribed for aniridia-associated keratopathy (AAK).

At the age of 14 months old, the patient, with vomiting, diarrhea, and subfebrile temperature, was referred local hospital. Clinical and radiological examination revealed a big right kidney tumor (Wilms tumor) with left kidney lesions suspicious of nephroblastomatosis. The patient was transferred to the Dmitry Rogachev National Medical Research Center of Pediatric Hematology, Oncology and Immunology for diagnosis and medical care.

Chemotherapy, according to the SIOP-RTSG 2016 Umbrella protocol, was started, a regimen for bilateral disease with predisposition syndrome. The patient underwent 6 weeks of AV without a good response. An MRI showed a 20% increase in right kidney tumor size. The therapy scheme was switched and intensified with one course of VP16/Carbo, but with no good response either; on the 11th day of the course, the abdominal circumference increased to 55 cm. According to the CT, there was a 40% increase in the right kidney tumor, and left kidney lesions seemed to be without dynamics. 

The lack of response to chemotherapy might have indicated a histological stromal type of Wilms tumor. Thus, a decision was made to perform surgery instead of chemotherapy prolongation. 

Transverse laparotomy and both kidneys NSS were performed. Histology showed WT, stromal type, intermediate risk, local stage II (renal sinus invasion) in the right kidney, and nephroblastomatosis lesions in the left one. 

Adjuvant chemotherapy, in this case, was performed according to the AV-2 scheme (27 weeks), and maintained chemotherapy for nephroblastomatosis was given monthly up to one year from the start of neoadjuvant treatment.

The child achieved a full response and was left under observation with a follow-up duration of 37 months from the start of the treatment.

Because of a congenital penis defect at the age of 18 months, orthoplasty of the penis, Bracka two-stage repair surgery, was carried out. Functional gonadal tissue was defined. The patient suffers from secondary obstructive pyelonephritis, frequent relapses, incomplete doubling of the right ureter, and secondary obstructive ureterohydronephrosis of the doubled segment of the right kidney.

At the age of three years, the patient’s ocular manifestations included congenital complete aniridia, nystagmus, fovea hypoplasia, drug-compensated glaucoma, aniridia-associated keratopathy (stage 1), ptosis (degree 1), and high myopia (Figure 1).

The patient has a mild delay in speech and motor development and no other neurological disorders.

### 2.2. Genetic Findings

At the age of 2 years old, the patient was referred to the Research Centre for Medical Genetics, where WAGR syndrome was diagnosed. In order to confirm the diagnosis, the patient underwent a series of genetic testing.

During a standard cytogenetic analysis, abnormal male karyotype 46,XY, accompanied by complex rearrangement, specifically reciprocal translocation between short arms of chromosomes 10 and 11 t(10,11)(p15;p13), was revealed. A pericentric inversion inv(11)(p13q12) was also suspected but required additional investigation (Figure 2). Derivative chromosome 11 appeared to be nearly acrocentric.

Region-specific FISH analysis for the short and long arms of chromosome 11 demonstrated that a fragment of the short arm of derivative chromosome 11 appeared to be situated under the centromere on the long arm of this derivative der(11) chromosome. These findings confirm the additional pericentric inversion of derivative chromosome 11, inv(11)(p13q12). Additional mBAND confirmed the results of the FISH analysis (Figure 3). The analysis of the hybridization profiles of the abnormal chromosomes confirms pericentric inversion of derivative chromosome 11 (Cy5 signal replaced after centromeric region) and that the false color showed produced in the terminal region of der(10) is true chromosome 11 signal. According to the performed analysis, the patient’s karyotype is 46,XY,der(10)t(10;11)(p15;p13),der(11)inv(11)(p12q12)t(10;11)dn.

Furthermore, to determine the possible presence of cryptic imbalances that occurred during chromosome rearrangements, chromosomal microarray analysis (CMA) was performed (Figure 4). The analysis showed two cryptic interstitial microdeletions associated with the breakpoints on chromosome 11: arr [hg19] 11p14.1p13(30440294_32696663)x1, 11p13p12(32772515_36857171)x1. The deletion located in the region 11p14.1p13 included, among others, the *WT1* (MIM #607102) and *PAX6* (MIM #607108) genes, which correlated with the clinical phenotype of WAGR syndrome in the patient.

## 3. Discussion

The patient with complex chromosomal rearrangements and local imbalance at the breakpoints of chromosome 11p13 presented all four classical features of WAGR syndrome: early developed bilateral renal tumor, congenital aniridia, genitourinary anomalies, and intellectual disability. He was born with a birth weight of 4.8 kg (median is 2.9 kg), and he was diagnosed with Wilms tumor at the age of 15 months, which is very early since the median age of WAGR-associated Wilms tumor diagnosis is 22 months [5].

In the neonatal period and early infancy, patients with isolated aniridia and with WAGR syndrome are clinically undistinguishable [4]. Indeed, we reveal WAGR patients among patients with congenital aniridia during molecular diagnostics before Willms tumor development [6]. This poses the necessity of early referral for such patients to a clinical geneticist in order to differentiate WARG syndrome and congenital aniridia with syndromic features.

Nevertheless, the patient described here was first referred to the oncologist for tumor treatment. In this case, nephroblastoma developed very early. Regular oncological surveillance was conducted, which showed complete remission with no sign of relapse, with secondary obstructive hydronephrosis on the right.

In the patient, congenital aniridia combined with nystagmus, glaucoma, optic nerve hypoplasia, and fovea hypoplasia are the most common features of the disease [7,8]. Ocular hypertension and glaucoma occur in 25–50% of cases of WAGR syndrome, and optic nerve and fovea hypoplasia are common features of an aniridic eye. Corneal abnormalities, central corneal thickness, corneal vascularization, and opacity also occur in a substantial portion of the patients. Due to the high post-surgery risk of aniridia-associated keratopathy and/or fibrosis, glaucoma treatment in WAGR patients is difficult. Only about half of cases respond to conservative therapy [9]. Nevertheless, our patient’s intraocular pressure has been normalized after lasting conservative glaucoma treatment.

Under the supervision of various specialists, neurologists, psychologists, and special teachers, the boy, who is now 4 years old, attends a pre-school with a special educational program. We may conclude that the patient had a relatively mild course of the disease except for the early development of Wilms tumor, which nevertheless was successfully treated.

It was established that chromosome 11p13 imbalance occurred two times. It is interesting that the only imbalance that occurs during this complex chromosomal rearrangement is deletions on the 11p region. Chromosome 11 is known as a hot spot for breakage in chromosomal rearrangements [10]. The first upstream deletion in the region 11p14.1p13 encompassing *PAX6* and *WT1* gene was derived due to the reciprocal translocation between the short arms of chromosomes 10 and 11. Nevertheless, according to our data, this does not always lead to the development of WARG syndrome [11]. Most patients with so-called WARG deletions develop Wilms tumor only in cases when the *LMO2* gene is affected by deletion. The first upstream deletion in the patient does not affect the LMO2 gene, so we could hypothesize his chances of developing a Wilms tumor with only that deletion were low. Nevertheless, according to the proposed molecular mechanism, a second rearrangement happened, specifically pericentric inversion in derivative chromosome 11, inv(11)(p13q12), resulting in a second downstream contiguous deletion 11p13p12 located adjacent to the first one. This deletion disrupted *the LMO2 gene and was associated with a significant increase in* the risk for Wilms tumor development [11]. Figure 5 depicts the multistep mechanism of complex chromosomal rearrangement revealed in the patient.

A sufficient role in the patient’s phenotype is played by *PAX6* haploinsufficiency. It is a key regulator of eye development. It is required to initiate developmental pathways, execute specification and differentiation programs, integrate information, synchronize processes in different eye tissues, and interact with other regulatory pathways [12]. *PAX6* also plays an important role in brain development [13]. *PAX6* function deficit due to the deletion could contribute to WAGR patients’ phenotype in several ways. The deficit leads to eye affections either in the shape of classical aniridia complex or aniridia-like phenotype when iris hypoplasia and other eye anterior segment abnormalities combine with foveal hypoplasia. *PAX6* function deficit could also contribute to the neurological phenotype of WAGR patients and their brain integrity [14,15].

## 4. Materials and Methods

### 4.1. Clinical Methods

The diagnosis of congenital aniridia was based on the clinical presentation, including absence of the iris, foveal hypoplasia, and nystagmus. The severity of AAK was staged as defined by Holland et al. [16]. Stage I consisted of abnormal peripheral corneal epithelium, manifested by increased uptake of fluorescein (late staining). In addition to the standard ophthalmologic examination, patient underwent spectral-domain optical coherence tomography (SD-OCT) and gonioscopy.

### 4.2. Karyotyping

Cytogenetic analysis of the patient and her parents was performed on GTG-banded metaphase spreads obtained from cultured peripheral blood lymphocytes according to standard procedures. GTG-banded metaphase chromosomes were analyzed using Axio Imager A1 microscope (Carl Zeiss, Jena, Germany) with Ikaros Karyotyping System Software, V.5.8.14 (Metasystems, Altlussheim, Germany). The results were presented according to the International System for Human Cytogenomic Nomenclature 2020 [17].

### 4.3. FISH and Multicolor Banding

FISH was carried out using chromosomal preparations from cultured peripheral blood lymphocytes following the manufacturers’ protocols. DNA probes were used for subtelomeric regions of the short arm of chromosome 10 and the short arm of chromosome 11 (Sub-telomere 10pter, Sub-telomere 11pter, KREATECH, Amsterdam, The Netherlands); whole-chromosome probe was used for chromosome 10, the short arm, and the long arm of chromosome 11 (Whole Chromosome 10; Arm Specific short Probe 11; Arm Specific long Probe 11; KREATECH, Amsterdam, The Netherlands); and Satellite Enumeration Probe was used for chromosome 11 (CEP 11-SE 11 (D11Z1), KREATECH, Amsterdam, The Netherlands). Multicolor Banding DNA probe on chromosome 11 (XCyte Human mBAND probe; MetaSystems, Altlussheim, Germany) was applied. FISH results were analyzed using an AxioImager M.1 epifluorescence microscope (Carl Zeiss, Jena, Germany) and Isis digital image processing software (MetaSystems, Altlussheim, Germany). DNA probes for the chromosome 11-WT1/CEN11p (Abnova, Taipei, Taiwan) were applied. FISH results were analyzed using an AxioImager M.1 epifluorescence microscope (Carl Zeiss, Jena, Germany) and Isis digital image processing software (MetaSystems, Altlussheim, Germany).

### 4.4. Chromosomal Microarray Analysis (CMA)

The CytoScan HD array (Affymetrix, Santa Clara, CA, USA) containing 2.67 million markers using the GeneChip™ 3000 system (Thermo Fisher Scientific Inc., Waltham, MA, USA) was applied to detect the CNV across the entire genome, following the manufacturer’s protocols. Microarray-based copy number analysis was performed using the Chromosome Analysis Suite software version 4.0 (Thermo Fisher Scientific Inc., Waltham, MA, USA), and the results were presented on the International System for Human Cytogenomic Nomenclature 2020 (ISCN, 2020). Detected CNVs were totally assessed by comparing them with published literature and the public databases: Database of Genomic Variants (DGV) (http://dgv.tcag.ca/dgv/app/home, accessed on 9 September 2021), DECIPHER (http://decipher.sanger.ac.uk/, accessed on 9 September 2021), and OMIM (http://www.ncbi.nlm.nih.gov/omim, accessed on 9 September 2021). Genomic positions refer to the Human Genome February 2009 assembly (GRCh37/hg19). The pathogenicity of variants was evaluated according to the American College of Medical Genetics (ACMG) standard guidelines [18]. DNA samples were analyzed using chromosomal microarray analysis (CMA).

## 5. Conclusions

The clinical case highlights the need for numerous further studies to develop a common strategy for diagnosis and treatment of this nosology. Wilms tumor should be rapidly revealed in patients with aniridia as a symptom to prevent life-threatening complications. Our study also emphasizes the actuality of cytogenomic approaches in a postgenomic era.

## Figures and Tables

**Figure 1 ijms-24-16923-f001:**
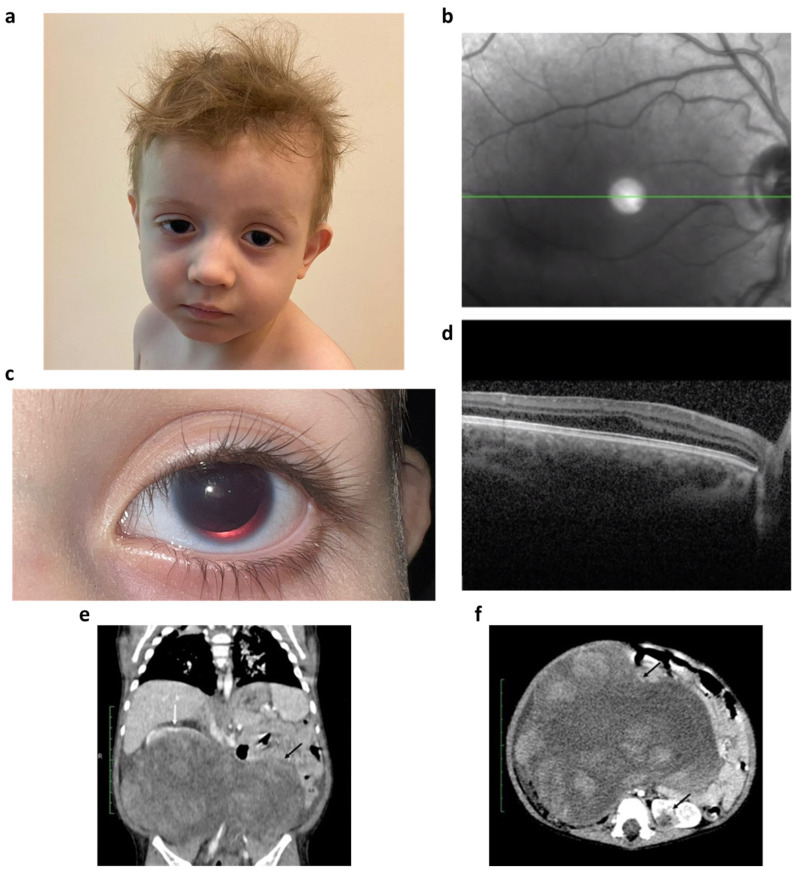
Clinical examination of the patient: (**a**) facial phenotype of the patient; (**b**) fundus imaging by OCT; (**c**) complete aniridia. Positive dynamics of glaucoma treatment for 4 years of the patient’s life was shown. We managed to avoid the progression of aniridia keratopathy and a reduction in corneal size; (**d**) spectral-domain optical coherence tomography (SD-OCT) in patient (OS). Hyporeflective fovea and macular hypopigmentation are revealed. Extrusion of plexiform layers and foveal pit are absent. Frontal (**e**) and transversal (**f**) contrast-enhanced CT sections of bilateral WT (black arrows) with normal kidney rim (white arrow).

**Figure 2 ijms-24-16923-f002:**
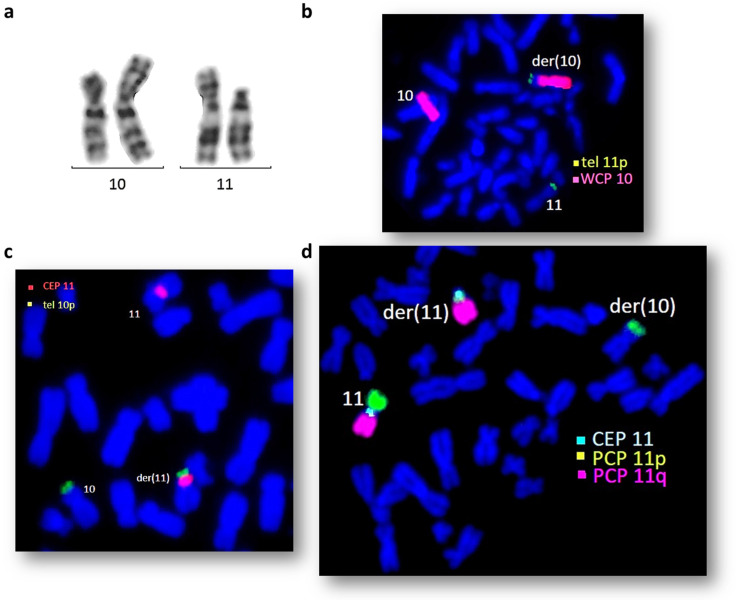
(**a**) Normal and derivative chromosomes 10 and 11 in the patient’s karyotype; (**b**) FISH with whole-chromosome 10 (red) and subtelomeric 11p (green) DNA probes; (**c**) FISH with DNA probes for centromere of chromosome 11 (red) and subtelomeric region of 10p (green); (**d**) FISH with DNA probes for 11p (green) and 11q (red) chromosomes and centromeric region of chromosome 11 (blue).

**Figure 3 ijms-24-16923-f003:**
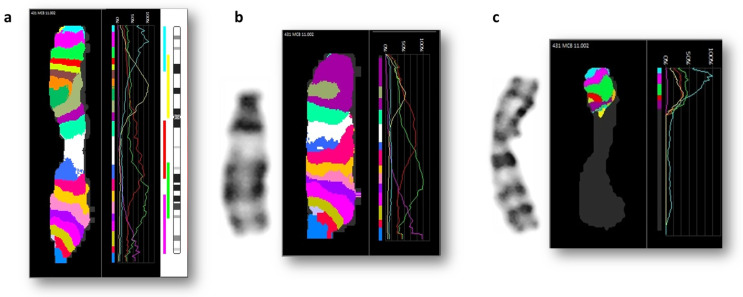
Multicolor banding (MCB) analysis are evaluated by hybridization profiles (signal intensity relative to the strongest signal on this chromosome) of corresponding fluorochromes: FITC, SpO, TR, Cy5, DEAC. (**a**) Normal homologue of chromosome 11; (**b**) Derivative chromosome 11; (**c**) Derivative chromosome 10.

**Figure 4 ijms-24-16923-f004:**
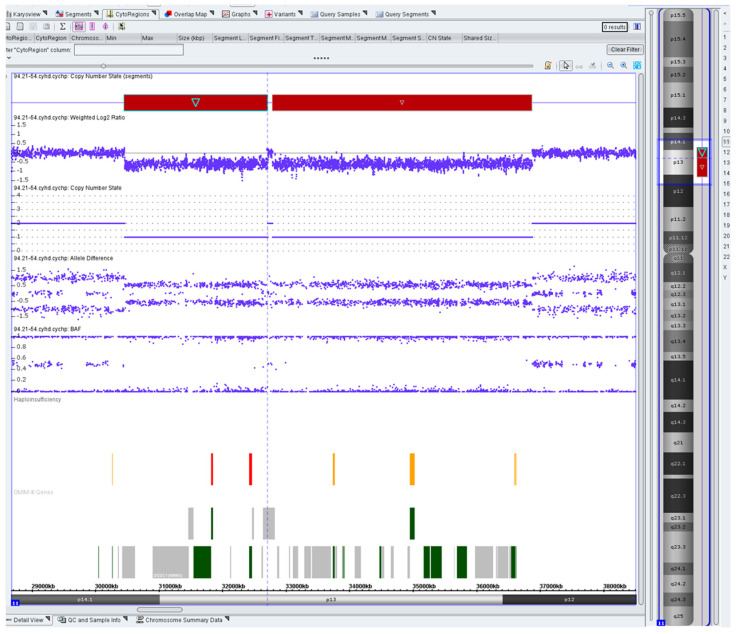
Chromosomal microarray analysis revealed 2 adjacent contiguous deletions: arr [hg19] 11p14.1p13(30440294_32696663)x1 and 11p13p12(32772515_36857171)x1 at the breakpoints of complex rearrangement.

**Figure 5 ijms-24-16923-f005:**
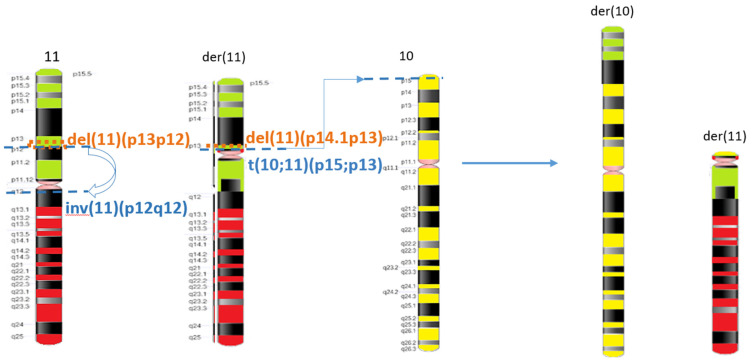
Proposed mechanism of formation of the complex chromosomal rearrangement. Schematic representation of the chromosomes involved in the complex chromosomal rearrangement shows their breakpoints (blue lines) and two microdeletions detected by CMA (orange strokes).

## Data Availability

The datasets used and/or analyzed during the current study are available from the corresponding author upon reasonable request.

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
