# Peer review of "Complex Chromosomal Rearrangement Involving Chromosomes 10 and 11, Accompanied by Two Adjacent 11p14.1p13 and 11p13p12 Deletions, Identified in a Patient with WAGR Syndrome"

_ijms, 2023, doi:10.3390/ijms242316923_

Round 1

Reviewer 1 Report

Comments and Suggestions for Authors

This study is a case report that identified the mutant site through the Congenital disorder case and Chromosomal Microarray Analysis identified chromosomal re-arrangements of chromosomes 10 and 11.

The authors predicts that the causal gene as the PAX1 which was reported in other similar cases.

Overall, it is meaningful as a case report, but the academic interest is not high.

Comment 1. Add more explanation on the role and mechanism of the PAX1 gene for the developmental process

Author Response

This study is a case report that identified the mutant site through the Congenital disorder case and Chromosomal Microarray Analysis identified chromosomal re-arrangements of chromosomes 10 and 11.

The authors predicts that the causal gene as the PAX1 which was reported in other similar cases.

Overall, it is meaningful as a case report, but the academic interest is not high.

Comment 1. Add more explanation on the role and mechanism of the PAX1 gene for the developmental process

Answer 1. We have enriched the discussion by delving into the involvement of the PAX6 gene, specifically noting that its impact stems from deletion (notably, PAX1 located at 20p11 was not affected). This addition provides a more comprehensive understanding of the role of PAX6 in the broader development of the neural system, with a specific focus on its relevance to WARG syndrome.

Reviewer 2 Report

Comments and Suggestions for Authors

The authors present a complex case of WAGR syndrome with an unusual rearrangement between chromosome 11 and 10.  The workup using rigorous cytogenetic analysis is robust, and articulates the importance of a thorough chromosomal evaluation in these cases.

It also articulates the use of cytogenetics which these days is lost in the hype of whole genome sequencing.  The authors can emphasize how their classical approach still holds true today's molecular era.

Author Response

The authors present a complex case of WAGR syndrome with an unusual rearrangement between chromosome 11 and 10.  The workup using rigorous cytogenetic analysis is robust, and articulates the importance of a thorough chromosomal evaluation in these cases.

It also articulates the use of cytogenetics which these days is lost in the hype of whole genome sequencing.  The authors can emphasize how their classical approach still holds true today's molecular era.

Reply: Thank you for your insightful comment. We have taken your feedback into consideration and enhanced the conclusion section by incorporating this valuable point.